# An Experimental Study Evaluating the Influence of Front-of-Package Warning Labels on Adolescent’s Purchase Intention of Processed Food Products

**DOI:** 10.3390/ijerph19031094

**Published:** 2022-01-19

**Authors:** Lorena Saavedra-Garcia, Miguel Moscoso-Porras, Francisco Diez-Canseco

**Affiliations:** CRONICAS Center of Excellence in Chronic Diseases, Universidad Peruana Cayetano Heredia, Av. Armendariz 445, Miraflores, Lima 15074, Peru; miguelmporras@gmail.com (M.M.-P.); fdiezcanseco@gmail.com (F.D.-C.)

**Keywords:** nutrition labeling, front-of-pack labeling, warning labels, purchase intention, food preferences, adolescents

## Abstract

Front-of-package warning labels (WLs) are among the public health policies adopted by some countries, mainly in Latin-America, to tackle childhood obesity; however, their impact is still under review. The aim of this study was to assess, using an experimental design, whether WLs influence purchase intention of processed foods and identification of the healthiest products among adolescents in Peru, in May 2019, just before WLs mandatory implementation. Four hundred forty-nine adolescents from two public schools were randomly assigned to an experimental group (received three different processed products with either zero, one or two WLs, informing if they were high in sugar, sodium and/or saturated fats) or a control group (received the same products but unlabeled). Participants chose which product they would buy, and which they considered to be the healthiest. No differences between groups were found neither in purchase intention (*p* = 0.386) nor in the identification of the healthiest product (*p* = 0.322). In both groups, the most-selected product was always the healthiest of the triad (>40% in purchase intention and >80% in identification of the healthiest). Front-of-package WLs did not influence purchase intention, or identification of healthier products among adolescents from public schools in Peru. Mass media and educational campaigns should accompany the WLs implementation to help achieve the policy objective.

## 1. Introduction

In response to the rise in obesity prevalence among children and adolescents worldwide, several countries have adopted policies focused on reducing the consumption of processed and ultra-processed foods. This includes the taxation of sugar-sweetened beverages and marketing regulations on the advertisement of unhealthy foods, as well as the use of front-of-package labels (FOP) [1]. 

FOP are specific nutrient related information about a product presented as icons or schemes which are put front-of-pack of nutrition labeling [2]. They aim to help consumers make better-informed food choices in a quick and easy manner. There is a large variety of FOP schemes; for instance, some of them focus on specific nutrients such as sugar, saturated fats and sodium (e.g., warnings or “high in” symbols) and others present a summary label which includes a spectrum of criteria to establish product healthiness (e.g., Nutri-Score and Health Star Rated system) [3]. 

In the last decade, the FOP warning labels (WLs) scheme has become more common across the globe. Commonly, these WLs highlight the excess levels of critical nutrients (e.g., sugar, sodium, saturated and trans-fat) and recommend a lower intake of them. A conceptual model for how WLs affects consumers behavior has stated that for WLs to be effective, they must catch consumers’ attention and be accurately understood [4], and as they make decisions very quickly it is essential to use an eye-catching design. Then the WLs must elicit a perception of risk, which triggers behavioral intentions, and ultimately behavior change, and discourage consumers from purchasing “high-in” products, encouraging them to make healthier decisions. This model considers that external factors, such as preexisting values, nutritional knowledge and interpersonal communications, social reactions and norms, moderate the WLs effectiveness [5]. 

Various Latin American countries have adopted the use of WLs; for instance, Ecuador [6], Chile [7], Peru [8] and Uruguay [9] were the first in approving the regulation all with different labeling schemes. In June 2019 [8] despite a strong opposition from the food industry, Peru implemented its front-of-package WLs system, inspired by the black octagons introduced in Chile in 2016 [7]. The Peruvian octagons—four in total—indicate whether a processed product contains trans-fats (recommending avoidance of consumption) or if it is high in saturated fats, sodium or sugar (recommending avoidance of excessive consumption). The implementation of octagon WLs was a key component of a law focused on the promotion of healthy eating in children and adolescents, passed six years before in 2013 [10].

The available evidence on the impact of implementing mandatory WLs, although limited, shows positives changes in consumers’ eating behaviors. In Chile, one year after the implementation, mothers from children and adolescents from middle-upper socioeconomic level indicated paying attention and using labels in the buying decision-making process [11]. Furthermore, in the same country, an evaluation of the consumption of beverages has demonstrated a decline in purchases of those with WLs across all households’ education levels [12].

In addition to those policy evaluations, experimental studies are assessing the effects of different designs of WLs on the consumer’s choices or their ability to identify the healthiest processed foods. Results have indicated that aspects such as the color, shape and placement of the label have an impact on the perceived healthiness of a product [13].

Most of the studies aimed to assess WLs effectiveness have been conducted with adults, limiting the understanding on children and adolescents’ dietary behavior. A Brazilian study found that the octagon-shaped WLs modify healthfulness perception of products among children from 9–12 years [14]. Similar results were found in Uruguay among students from primary schools, where the impact of WLs was higher compared to the traffic-light system [15]. Therefore, more research is necessary to evaluate the impact of WLs in early ages, being that norms are focused in these life stages, and especially in adolescence, a critical period marked by the development of greater autonomy (for instance, they make more independent food purchase decisions compared to children [16]) that is considered a determinant for establishing health behaviors that influence health throughout the course of one’s life [17]. Furthermore, as food choice in young people is often influenced by product advertising, role models in media [18,19], or by parental influences [20], it is difficult to determine how much of the isolated effect of the WLs comes from proper understanding of healthy eating.

To date, most studies in this field have been developed online and have focused on comparing different WLs developed by researchers for their own studies, or tested effectiveness using labeling schemes implemented by foreign countries [21,22,23]. Natural experiments are necessary to analyze the real-world impact of the implementation of mandatory front-of-package WLs policies [5]. Therefore, we conducted an experiment to assess the influence of front-of-package WLs on the purchase intention and identification of the healthiest processed foods among adolescents, in May 2019, just before the implementation of mandatory WLs in Peru.

## 2. Materials and Methods

### 2.1. Design and Setting

Two-group randomized experiment (1:1) conducted in two public schools from Lima, Peru. We sent invitations to school’s principals from settings of different socioeconomic status; however, only two schools located in lower middle-income districts [24] agreed to participate in the experiment. Both schools offer primary and secondary education levels and had between 1300 and 1400 students each in 2019.

### 2.2. Participants

We recruited female and male adolescents between the ages of 10–14 years old who assented to participate voluntarily after receiving informed consent from their parents. Students of this age were selected since in similar contexts, studies have reported they are more independent when making decisions about food in comparison to younger children. For instance, they received money from their parents to buy snacks from school cafeterias [16]. Participants with visual impairments who were unable to read or distinguish the WLs messages were excluded. Based on a previous study performed by Khandpur et al. [21], sample size was calculated expecting to find a 16.3% difference in food choice between groups. This led us to recruit a target sample of 372 participants (186 per group) in order to have a statistical power of 90%.

### 2.3. Intervention

Students were visited at their schools during class hours and were invited to participate in the study by the project coordinator, who explained them that the study objective was to learn about adolescents’ food preferences. The front-of-package WLs were not mentioned to the principals, teachers, parents nor students at any point before the experiment took place, in order to prevent participants from reviewing on the topic, therefore influencing their responses. The team solved the students’ doubts about the experiment procedures. Informed consent forms were delivered to 763 students (463 and 300 in each school, respectively) to ask for parents’ permission, and only those who brought them back signed and provided their assent were included in the experiment.

The library of each school was adapted to receive the participants and isolate them from noise or other distractions during class hours. The experiment was carried out by five previously trained nutritionists, who individually interviewed each student and asked them whether they would like to answer some questions about food preferences and if they preferred sweet cookies or savory snacks. Subsequently, group allocation was performed through block randomization within each school. Interviewers opened a sealed envelope to know which group the participants would be assigned in.

Then, based on participant’s choice, the interviewer showed three different processed products that were previously stored and hidden from participants, and asked them two questions: (1) “Assuming all these products cost the same and are available at your nearest store, which one would you buy?” and (2) “Which of these three products you think is the healthiest?”. After answering each of them, interviewers made two open-ended questions to participants to explore the reasons for each of their choices.

The three processed products offered in the experimental group were labeled with zero, one and two WLs (octagons), while the control group received the exact same three products but unlabeled (Figure 1).

After participants made their choices, both groups were surveyed about their buying habits with a short, close-ended questionnaire. Additionally, the experimental group participants were asked whether they noticed something different in the products shown to them in order to assess if they noted the presence of the WLs. Once the exercise was completed, each participant received a stationery notebook as a present for their participation.

### 2.4. Product Selection

The six processed products used for the experiment were selected among the most sold in public-school canteens, according to five canteen owners interviewed during the design of the intervention. A pilot study was performed to test how familiar the children were with the selected products and to assess the questionnaire. All the selected products had similar prices (around USD 0.55) and package sizes within categories. The sweet cookies used were vanilla cookies, plain cookies and chocolate-filled cookies, and the savory snacks were popcorn, fried tortillas and puffed corn snacks. All products were single-serve packages (Figure 1).

As the experiment was performed one month before the implementation of WLs in the country, we designed the WLs (octagon stickers) with shape, color and dimensions according to the oncoming norm and placed them on the products of the experimental group (Figure 1). Products received the octagonal labels “high in sugar”, “high in sodium” and “high in saturated fats” which also followed the oncoming norm thresholds (nutritional composition of products is available in Appendix A) [8]. During the first 12 months of the WLs implementation, the use of stickers was permitted by the Peruvian government.

### 2.5. Outcomes and Analysis

The main outcomes were the difference in the proportions of purchase intention as well as the difference in the proportion of correct identification of the healthiest product between the experimental and control groups. Chi-squared tests were used to determine if groups were balanced before the experiment in terms of age, sex, current grade, food preference and buying habits, and to test differences in the dependent variables among groups. Purchase intention and identification data were analyzed separately for each food category, school, school level and sex. Finally, we performed a conditional logit regression analysis to assess differences by number of WLs between the experimental and control group.

Reasons given for each of the two product choices were classified independently by two researchers, in categories created after reviewing the qualitative answers. The discrepancies between the researchers were discussed to obtain agreement and the most common categories of reasons were plotted in bar charts (See Figure 2). Statistical analyses were performed using STATA V.15 (StataCorp, College Station, TX, USA).

## 3. Results

In total, 449 students participated in the study (225 in the experimental group). Randomization balanced (*p* ≥ 0.05) participants in terms of sex, age, buying habits and other relevant variables (Table 1).

No differences were found between the control and experimental groups in the two main outcomes. Most participants in both groups selected the vanilla cookies or popcorn (thus, the product with zero octagons) as the product they would buy among the three alternatives offered (46.9 vs. 44.9%, respectively, *p* = 0.386). Similarly, the vast majority of the experimental and control groups adequately identified the vanilla cookies or popcorn as the healthiest product of the trio (83.9 vs. 83.5% respectively; *p* = 0.322). These results were similar when we separately analyzed subgroups of food category (Table 2), school, school level and sex (see Appendix A). Additionally, in the conditional logit model, no difference was found between the control and experimental groups even when we compared the intention to purchase and identification of healthiest foods by number of labels (0 vs. 2 WL or 1 vs. 2 WL) (Appendix A).

When we asked for the reasons behind participants’ purchase intentions, around 75% of the responses in both groups were related to product taste, (e.g., “I like its taste”, “It has a delicious taste”), followed by nutritional composition related questions (e.g., “it has less sugar)”, habits (e.g., “I usually eat that”) and health benefits (e.g., “Is good for me”). Other reasons were less mentioned (see Figure 2).

The most common reasons to identify the healthiest product were linked to its nutritional composition. Around 40% in both groups mentioned reasons related to calories or critical nutrients (e.g., “Because the other (food) has calories and fat”, “Because it doesn’t have much sugar and that is good for us”), and around 30% in both groups explained their decision based on the ingredients (e.g., “Because it has cheese”, “Because it doesn’t have too much chocolate”). The degree of processing was mentioned, nevertheless, it was referred only in 8% of the control group and 5.8% in the experimental group (see Figure 2).

In the experimental group, the presence of the WLs was hardly mentioned as a reason to explain the purchase intention and only 7.6% mentioned them to justify their healthiest product choice. Finally, when asked, 68.0% (*n* = 153) of students in the experimental group answered that they had noted the WLs or its message in the products, 24.4% did not identify something different in the packages and 7.6% noted elements different from WLs (i.e., cartoon or claim).

## 4. Discussion

Our study did not find differences in the food purchase intentions nor in the ability to identify the healthiest processed foods between low-income adolescents who received products with and without front-of packages WLs.

It is possible that in our study setting, the sole presence of front-of-package WLs may be insufficient to influence purchase intentions of adolescents from lower middle-income communities. Two results that support this hypothesis are that the octagons were hardly mentioned by students from the experimental group as the reason for their purchase intention, and that 32% of the same group did not mention the octagons as a new feature of the products’ packages. It is relevant for WLs to catch consumers’ attention as they make quick decisions, but at the same time to inform and finally orient consumers decisions [4]. Given that this study was developed before any public educational or communicational campaign was launched to informed about WLs implementation, there was a lack of information and motivation that can trigger behavioral response to choose the healthiest product [5,25].

Our results differ from other similar experimental studies, including some with children and adolescents, that have found that WLs usually dissuade consumers from purchasing foods high in critical nutrients and help in the identification of the healthiest products [5]. For example, a study with adolescents (16–18 years) in New Zealand found that the presence of octagon WLs significantly reduced the intention to purchase breakfast cereals of low nutritional value [26]. In Uruguay, a study involving children from 8 to 13 years found that two designs of WLs (the traffic light system and octagons) reduced the choice of products high in critical nutrients [15].

However, there is also evidence that the consumers’ characteristics, such as their socioeconomic status and age, could influence the purchase decisions and reduce the influence of the WLs. For example, in a Brazilian study, the presence of the WLs helped identifying how healthy a product was among 9- to 12-year-old students from private schools, but not in the group from public schools, as in our study. Additionally, in the same study, WLs did not influence the identification of the healthiest products in the youngest group of participants aged between 6 and 8 years old, regardless of whether the school they attended was public or private [14]. Moreover, other studies have reported limited effects of WLs. For instance, in a Canadian study, researchers found no effect of different schemes of WLs in the intention to purchase a sugary beverage in consumers over 16 years. Authors explained consumers’ response due to the exercise was carried out in conditions with no previous information about the WLs [25].

According to the majority of students in both of our study groups (see Figure 2), the main reason for selecting a product was its taste. In addition, the proportion of students that correctly recognized the healthy product in each group (four out of five) was clearly superior to the proportion of students that select the healthiest (one out of two), which means that a group of students, while they recognize the healthiest products, do not choose them. This result is aligned with other investigations that suggest that taste preferences, rather than nutritional information or health benefits [27], are the strongest influencers in consumers’ buying decisions, especially among children and adolescents. Interestingly, in our study, the preferred product was always the healthiest one (vanilla cookies or popcorn), something that could suggest that the adolescents’ choices are not necessarily the least healthy.

When we asked for the healthiest product, results were similar to the first election and most participants correctly identified vanilla cookies or popcorn. It is possible that this “healthiness” can be easily differentiated independently of the presence of WLs as vanilla or corn are ingredients with a natural origin or are commonly found in home in comparison to chocolate or puffed corn. This idea coincides with the answers some students (around 20%) provided in the open-ended questions where they addressed that the healthiest products came from “natural sources”.

The absence of improvement in the identification of healthiest products could be explained by actual evidence that shows that WLs may have greater potential to discourage consumers from making unhealthy food choices rather than highlighting healthy products. Due to WLs highlight “high-in” critical nutrients, they communicate the idea that products are not healthful [14]. In addition, studies demonstrates that FOP tend to only influence healthfulness perception of products that are wrongfully perceived as healthful, whereas they did not change consumer perception in the case of products that were previously clearly identified as healthy or unhealthy (e.g., potato chips) [28,29].

### 4.1. Strenghts and Limitations

A strength of this study is that was performed one month before the use of WLs became compulsory in Peru, when only some products had introduced octagons and no educational or media campaigns had been implemented. This enabled the assessment of the solely influence of WLs in participants who have little or no familiarity with them and had no additional information about their meaning.

The limitations of the present study should be acknowledged. First, generalizability of findings is limited due to participants being selected from two public schools of lower middle-income communities. Second, products considered in the study were real products that included elements in the label design (e.g., cartoons, health claims) that we did not evaluate and could influence consumer’s decisions and pre-existing attitudes and knowledge towards products could influence their decisions. However, using products currently available in the market allowed us to portray the scenario when WLs enter into force. Third, as students could talk to each other about the exercise, there is a likelihood of a sample contamination. Nevertheless, participants were not informed about the real objective of the study, and the exercise was carried out during class hours in only 2 days, the first day for primary and the latter for secondary, to minimize the possibility that students talked about the experiment during recess or at the end of the day.

### 4.2. Public Health Implications

The study findings could be of interest for policy makers who need evidence to support and improve the implementation of WLs policies in favor of healthy diets and health promotion among children and adolescents.

The front-of-package WLs are implemented as a policy to provide standardized information to consumers and motivate better food choices. In our study, performed just before their mandatory implementation in Peru, the front-of-package WLs did not seem to influence the purchase intention, nor the identification of the healthiest processed products among low-income adolescents. Importantly, the Peruvian government media campaigns regarding the introduction of the WLs were weak and only started after their implementation. Additionally, it was not until 2020, together with the challenges of the COVID-19 pandemic, that the WLs were included in the schools’ curricula.

The results invited identification and analysis of other elements needed to drive behavior change in adolescents from low-income communities to make better decisions. To reach the WLs policy aims focused on children and adolescents, their implementation should be supported by other actions such as simultaneous communicational and educational campaigns, to increase the awareness of their presence and reinforce their use and message [16]. In Uruguay, a recent study showed the need to create different types of communication campaign messages according to gender, age and nutritional status of the target population when promoting the use of WLs [30], showing that diverse approaches are necessary in the population segments since each one responds differently to the WLs. Informing and educating children and adolescents to adequately use the WLs when choosing their food is essential, especially when ultra-processed products are highly available in school environments [31].

This study only analyzed the influence of WLs in two food categories; however, for future studies it is relevant to evaluate the impact in more food categories, in real scenarios (e.g., school cafeterias) and in other countries where WLs have already been implemented. Additionally, studies should consider potential external factors, social reactions and norms that would modify the labels’ effectiveness on healthfulness perception and purchase intention to understand the real impact on behavioral change.

## 5. Conclusions

Front-of-package WLs did not influence the purchase intention, nor the identification of the healthiest processed products between low-income adolescents one month prior to the implementation of mandatory WLs in Peru. Further studies on how other features of the product, and individual and external factors such as familiarity, could influence WLs effectiveness in the real world are necessary in Peru and countries with WLs policy implemented. Simultaneous awareness and educational campaigns ought to be implemented to reinforce the use of the WLs and contribute to better food purchase.

## Figures and Tables

**Figure 1 ijerph-19-01094-f001:**
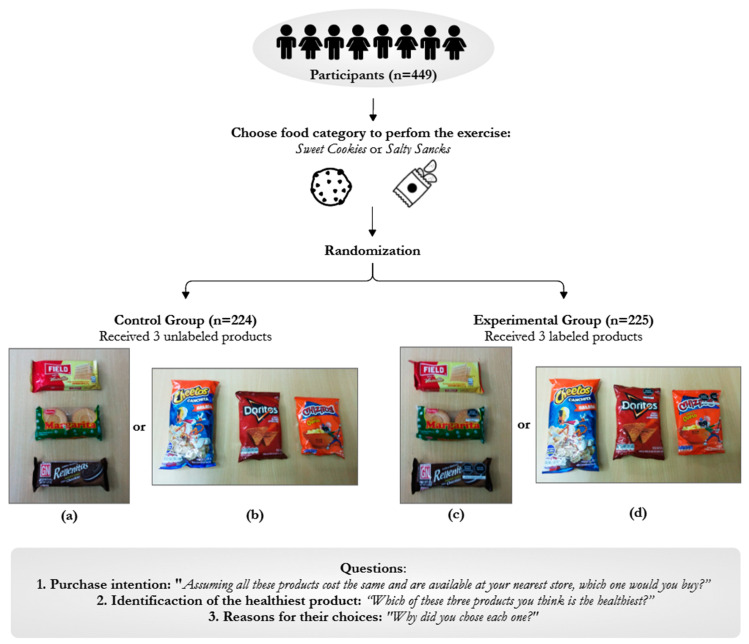
Distribution of participants and processed products offered. (**a**) Sweet cookies for control group (unlabeled): vanilla cookies, plain cookies, and chocolate-filled cookies; (**b**) savory snacks for control group (unlabeled): popcorn, fried tortillas, and a puffed corn snack; (**c**) sweet cookies for experimental group (labeled): vanilla cookies with no octagon, plain cookies with one octagon, and chocolate-filled cookies with two octagons; (**d**) savory snacks for control group (unlabeled): popcorn with no octagon, fried tortillas with one octagon and a puffed corn snack with two octagons.

**Figure 2 ijerph-19-01094-f002:**
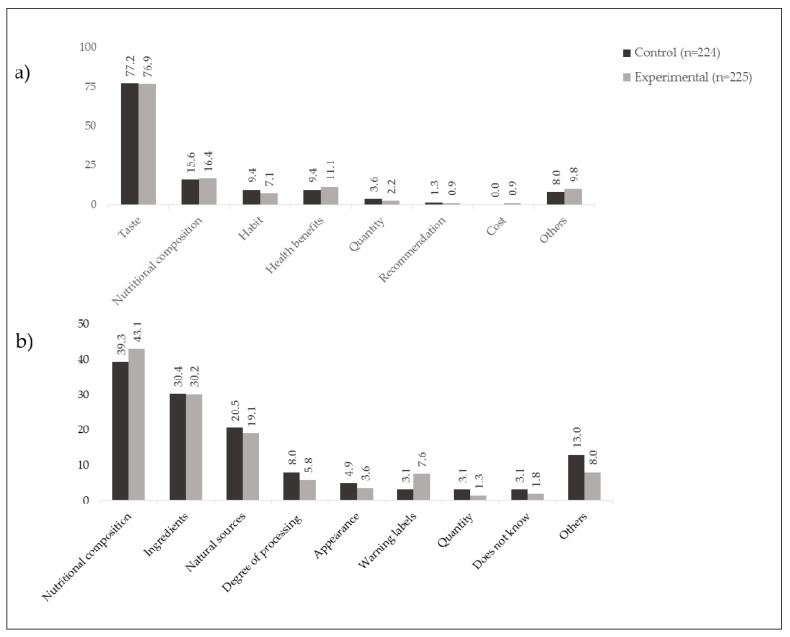
Reasons reported for (**a**) Purchase intention and (**b**) identification of healthiest product.

**Table 1 ijerph-19-01094-t001:** Participants’ characteristics per group.

Characteristics		Control * (*n* = 224)	Experimental (*n* = 225)	*p*
		*n* (%)	*n* (%)	
Gender (Female)		135 (60.3)	129 (57.3)	0.528
Age (Years)	10	43 (19.2)	35 (15.6)	0.342
	11	40 (17.9)	52 (23.1)	
	12	69 (30.8)	71 (31.6)	
	13	48 (21.4)	37 (16.4)	
	14	24 (10.7)	30 (13.3)	
Educational level	Primary	86 (38.4)	87 (38.7)	0.952
	Secondary	138 (61.6)	138 (61.3)	
Buys food on his/her own		196 (87.5)	198 (88.0)	0.872
Checks package information when buying food		144 (73.5)	149 (75.3)	0.685
Chosen food category for experiment	Cookies	142 (63.4)	151 (67.1)	0.408
	Snacks	82 (36.6)	74 (32.9)	

* Received products without warning labels on the package; WL: warning labels.

**Table 2 ijerph-19-01094-t002:** Differences in intention to purchase and identification.

	Intention to Purchase		Identification of Healthiest Food			
	Control *	Experimental	*p*	Control *	Experimental	*p*
	*n* (%)	*n* (%)		*n* (%)	*n* (%)	
**Cookies**			0.196			0.855
Vanilla cookies (0 WL)	67 (47.2)	65 (43.1)		115 (81.0)	126 (83.4)	
Plain cookies (1 WL)	41 (28.9)	58 (38.4)		24 (16.9)	22 (14.6)	
Chocolate-filled cookies (2 WL)	34 (23.9)	28 (18.5)		3 (2.1)	3 (2.0)	
**Savory Snacks**			0.950			0.113
Popcorn (0 WL)	38 (46.4)	36 (48.7)		73 (89.0)	62 (83.8)	
Fried tortillas (1 WL)	33 (40.2)	29 (39.2)		5 (6.1)	2 (2.7)	
Puffed corn snacks (2 WL)	11 (13.4)	9 (12.1)		4 (4.9)	10 (13.5)	
**All**			0.386			0.322
Vanilla cookies and popcorn (0 WL)	105 (46.9)	101 (44.9)		188 (83.9)	188 (83.5)	
Plain cookies and fried tortillas (1 WL)	74 (33.0)	87 (38.7)		29 (13.0)	24 (10.7)	
Chocolate filled cookies and puffed corn snack (2 WL)	45 (20.1)	37 (16.4)		7 (3.1)	13 (5.8)	

* Received products without warning labels on the package; WL: warning labels.

## Data Availability

The data presented in this study are available on request from the corresponding author.

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
