# Peer review of "An Experimental Study Evaluating the Influence of Front-of-Package Warning Labels on Adolescent’s Purchase Intention of Processed Food Products"

_ijerph, 2022, doi:10.3390/ijerph19031094_

Round 1

Reviewer 1 Report

The authors presented an interesting study on the influence of WL among adolescents. This study, despite the issues with generalizability, has a potential impact on policy related to WL / labelling. 

Overall, I find the introduction provided sufficient background and justification to the study. The objective and methods were clearly described. 

The data analysis was simple but it addressed the objectives. 

However, I believe the discussion is too brief has to be strengthened with more evidence from other parts of the world, especially now that the study did not find a difference between the experimental and control groups. 

The paper, otherwise, is well-presented. 

Author Response

Dear reviewer, 

Best regards, 

Reviewer 2 Report

The paper submitted deals with research aimed at identifying consumer choice in young people based on nutritional warning label information.
In itself, the topic of the study is interesting, as it aims to identify the importance attributed by adolescent consumers to nutritional labeling (front-labeling warning labels) on food snacks.

However, the authors report that the study was performed before the mandatory labeling of food packaging was introduced. Therefore, it was expectable that consumers were not interested or informed about the existence of the labels, a fact further confirmed also by the results that they preferred (following the open answers requested during the interviews) the product they considered healthier.
Thus, the statement that this lack of knowledge should be considered a strength is highly questionable, if not entirely wrong. The authors state that participants had "little or no familiarity with them and had no additional information about their meaning", therefore it is not clear how they could have assessed the importance of warning labeling at all.

Furthermore, the description of the method of statistical analysis of the interview results is very poor, limited to a chi-square evaluation of results otherwise limited to answer percentage distribution.

In the results and discussion section is stated that other similar studies conducted in countries where warning labels were introduced showed higher dissuasive power in choosing non-healthy food products, but reasons given to support the opposite outcome of the present study are limited to the suggestion that "the adolescents' choice are not necessarily the last healthy". Conclusions section, finally, is scarce and limited to a generic proposal for further investigations.

Thus, in the opinion of this reviewer, the paper does not contain sufficient elements to justify its publication in the journal. It is advisable to resubmit after repeating the study under informed conditions on a collective comparable to the one studied and to conduct an inferential analysis to evaluate the influence of warning labels in food choice.

Author Response

Dear reviewer, 

We appreciate the positive and constructive feedback provided. Attached you will find a point-by-point response to each comment.

Best regards,

Reviewer 3 Report

As the authors state “The aim of this study was to assess, using an experimental design, whether Front-of-package warning labels (WLs) influence purchase intention of processed foods and identification of the healthiest products among adolescents in Peru, just before WLs mandatory implementation.

-The paper presents an interesting topic. It contains new and significant information adequate to justify publication. Some comments are next given that help authors to improve the manuscript.

INTRODUCTION

-Some more definitions are required for the readers of the paper in regards the (food) octagons / octagon labels.

-The authors should make clearly apparent why it is important to study adolescents choices

They should sufficiently explain the research strategy they follow (experiment). It is not clearly apparent why they claim "comparing different WLs designed by researchers or using labeling schemes from different countries, then natural experiments are necessary to analyze the real-word impact of the implementation of mandatory front-of-package WLs in children".

MATERIALS AND METHODS

- Authors should theoretically define how the students' choices are linked with WLs perceptions/observations

- The authors should clearly also define why "Two-group randomized experiment (1:1)" have been chosen.

- Why they recruited female and male adolescents between the ages of 10-14 years?

RESULTS

- The authors should note why they use chi squared test instead of something other, for example a discriminant model?

- They should note the program they used to analyze the data.

- Think to add a third bar reflecting the differences between the two groups in figure 2.

DISCUSSION AND CONCLUSION SECTIONS

- Although some discussion is presented the authors could deeper discuss the results of the study comparing them with those of analogous studies.

- Although limitations due to the sample there exist, they could give some generalizations.

- Because there are no propositions for future research, it is recommended to be added one or two propositions.

Author Response

(The authors gave the same response as above.)

Round 2

Reviewer 2 Report

This reviewer went through a critical reading of the new version of the paper submitted, with particular attention given to the authors' responses in their cover letter.
After comparison with the first submission, the paper shows still limitations for methodological hypotheses, sample taken into consideration, and statistical data handling; however, the authors state clearly in the conclusions that the study needs more investigation and should be seen as a research starting point.
Moreover, the authors made their best effort to address all reviewer's comments: thus, the revised version of the manuscript shows substantial improvements, especially in the introduction and in the discussion section.
Therefore, as the authors in their answer held on their positions for statistical data handling, I will refrain from rephrasing the same comments in this answer to them. If the editor believes that the paper is sufficiently tailored for the scope of the journal, I have no other objections to its publication.

Author Response

1. This reviewer went through a critical reading of the new version of the paper submitted, with particular attention given to the authors' responses in their cover letter.
After comparison with the first submission, the paper shows still limitations for methodological hypotheses, sample taken into consideration, and statistical data handling; however, the authors state clearly in the conclusions that the study needs more investigation and should be seen as a research starting point.
Moreover, the authors made their best effort to address all reviewer's comments: thus, the revised version of the manuscript shows substantial improvements, especially in the introduction and in the discussion section. 

Response: We wish to express our gratitude to the reviewer for thetime invested in reading our manuscript. 

2. Therefore, as the authors in their answer held on their positions for statistical data handling, I will refrain from rephrasing the same comments in this answer to them. If the editor believes that the paper is sufficiently tailored for the scope of the journal, I have no other objections to its publication.

Response: We have incorporated a conditional logit regression analysis to assess differences by number of WLs between groups as part of the supplementary materials. Additionally, this modification is aligned with the editor request. 
